# Passive vs. active warm-up combined with stretching on hamstring flexibility and maximal voluntary contractions

Marion Hitier[1,2,3], Denis César Leite Vieira [1,2,4], Carole Cometti[1,2],
Joao Luiz Quagliotti Durigan[5], Nicolas Babault [1,2]*

1 Sport Science Faculty, Cognition, Action and sensorymotor plasticity (INSERM UMR1093-CAPS), Université Bourgogne Europe, Dijon, France, 2 Sport Science Faculty, Centre for Performance and Expertise, Université Bourgogne Europe, Dijon, France, 3 Chiropractor, Private Practice, Dijon, France, 4 Department of Physical Education, Graduate Program of Physical Education, Catholic University of Brasilia, Taguatinga, Brazil, 5 Laboratory of Muscle and Tendon Plasticity, Graduate Program in Rehabilitation Sciences, Universidade de Brasília, Brasília, DF, Brazil

* nicolas.babault@u-bourgogne.fr

## Abstract

This study aimed to compare the effects of passive and active warm-up protocols combined with static or neurodynamic stretching on hamstring muscle function. Sixteen individuals (7 men and 9 women) performed three experimental sessions in a randomized order: 1) passive warm-up and static stretching, 2) passive warm-up and neurodynamic stretching, 3) active warm-up and static stretching (control condition). Passive warm- up consisted of 20 minutes in a 45°C hot-room. Active warm-up included 10 minutes of cycling and 10 minutes of sub-maximal contractions. Following warm-up, the participants were engaged in six sets of 30-second stretches, either performed using static or neurodynamic modalities. Testing involved two maximal voluntary contractions (MVC), a passive knee extension test (to evaluate range of motion and hamstring stiffness), and a stand-and-reach test (used for flexibility assessment) conducted before, after warm-up, and after stretching. Electromyography from the biceps femoris and semitendinosus were recorded during MVC. Results revealed a significant time effect for flexibility ($p < 0.001$). Flexibility enhancements were obtained following active and passive warm-ups and further increased after the stretch, independently of the stretch intervention. The electromyographic activity of the semitendinosus muscle was affected by the time ($p = 0.004$). It revealed a decrease after stretching as compared to a post-warm-up measurement. No other differences were observed between conditions and time for maximal torque and stiffness indexes. It is concluded that both the active and passive warm-up methods are efficient to increase flexibility. Irrespective of the modality, stretching further improved flexibility without any alteration in muscle viscoelastic properties.

**Data availability statement:** All data files are available from a public database with the following DOI: 10.17632/3znpm68tyy.1 (https://data.mendeley.com/datasets/3znpm68tyy/1).

**Funding:** Région Bourgogne Franche-Comté (2020Y-22065 and 2022Y-13186) National Council for Scientific and Technological Development–CNPq (200391/2022-4).

**Competing interests:** The authors have declared that no competing interests exist.

## Introduction

Warm-up, a multi-component routine preceding most training and competitions, is commonly used to prepare individuals for subsequent efforts and to reduce injury risks [1]. It is composed of various situations to enhance numerous physiological functions and performance parameters [2]. For instance, increases in muscle force, sprint running velocity, blood circulation and joint range of motion are often documented [3–5].

Traditional warm-up routines are mainly active. It is generally composed of low-intensity activities such as running or cycling, flexibility, activity-specific exercises and could be associated with high-intensity contractions at the very end [6–8]. Physiological mechanisms mostly include increases in muscle and core temperature associated with multiple neural alterations such as increases in corticospinal excitability or conduction velocity [3,7,9]. However, passive warm-up strategies recently regain particular attention [3,10]. Passive strategies do not require any voluntary contractions. They are generally achieved using hot conditions such as clothes or warm rooms [11] although not exclusively. For instance, motor imagery can also be used [6]. In a very recent review considering the limited existing dataset available from the whole literature, authors stated that active and passive warm-up produced similar effects on force-time parameters [3]. Moreover, passive strategies have been shown to have the advantage to save energy and play a major role at a psychological level as compared to active ones [10,12]. Therefore, the composition of warm-up routines is today challenged.

Amongst warm-up activities, stretching is often used more particularly for flexibility or range of motion improvements [13]. However, because static stretching can have detrimental effects for force production (more particularly with durations longer than 60 seconds) [14–16]. Consequently, other forms of stretching, such as the dynamic stretching, are proposed to complement warm-up routines [17,18]. Some alternatives such as the neurodynamic stretching focusing on nerve tension relieving [19–21] could also be of interest in sports settings [22]. This unusual form of stretching is mostly used in clinical practices to restore the dynamic balance between the relative movement of neural tissues and the surrounding mechanical interfaces [19]. Indeed, as compared to static stretching, a greater increase in range of motion has been observed [23,24]. However, because this form of stretching mainly focuses on nerves, a lower force production alteration could be hypothesized but remain to be determined. The stretching modality to include inside a warm-up routine is therefore questionable.

In addition, recent studies suggested stretching do not provide supplementary effects for range of motion when included in warm-up routines [4]. These authors hypothesized that any activity aiming at increasing temperature could be used interchangeably or in combination to increase range of motion. Interestingly, improved viscoelastic properties resulting from increased temperature could be obtained either using active or passive warm-up [10,25]. Accordingly, to confirm the usefulness of stretching during warm-ups, the effects of warm-up activities on range of motion, whether active or passive, remain to be rigorously compared to stretching [4].

This study aimed to compare the active and passive warm-up routines combined with stretching on neuromuscular properties and flexibility or stiffness. We hypothesized that passive warm-up would similarly reduce stiffness and enhance flexibility than active warm-up. Additionally, we hypothesized that static stretching would have no additional effects on flexibility or stiffness than warm-ups while neurodynamic nerve gliding would lead to greater gains.

## Materials and methods

### Participants

Sixteen volunteers (7 women and 9 men) participated in this study. The mean age ± standard deviation (SD), height, and body mass of women were 21.7 ± 2.2 years, 165.0 ± 2.8 cm, and 57.6 ± 7.8 kg, respectively. For men, the mean age, height, and body mass were 0.8 ± 1.7 years, 179.5 ± 9.6 cm, and 70.9 ± 10.2 kg, respectively. All volunteers were competitive athletes from various sports (track and field, handball, and soccer) with an average of 7.7 ± 3.6 hours training per week. None of them reported any lower limb injury or back pain in the last three months nor any specific hamstrings injury in the last two years. During the present experiment, participants were asked to maintain their regular physical activities and dietary intake. They were also advised to refrain from intensive activity at least during the two days prior to an experimental session. All volunteers were fully informed about the experimental procedure and purpose of the study. They read and signed an informed consent form. The study was conducted following the Declaration of Helsinki and received approval from the Ethics Committee for Research in STAPS (CERSTAPS IRB00012476-2022-15-03-166). The sample size (n = 15) was calculated a priori using G*Power (version 3.1.9.6) based on an effect size of 0.40, power of 0.9, and probability error of 0.05. The recruitment started April the 1st 2022 and ended the June 30th 2022.

### Experimental design

This study was a cross-over, randomized, and single-blind trial. Blinding was used for data analyses and statistics (N.B.). The volunteers visited the laboratory on four occasions. The first visit was used as a familiarization for all tests, warm-up, and stretching procedures. Anthropometrics (height, body mass) and lower back and hamstrings flexibility (stand-and-reach test) were also evaluated. The subsequent three randomized experimental sessions were: passive warm-up followed by static stretching (PasSTA), passive warm-up followed by neurodynamic stretching (PasND) and control condition (CON) composed of active warm-up followed by static stretching (Fig 1). Participants were tested at three-time points during all experimental sessions: before (PRE), after warm-up (WARM), and after stretching procedures (STRETCH). Tests, focusing on the hamstring muscles, included: passive knee extension stretch until the maximal point of discomfort to evaluate range of motion,

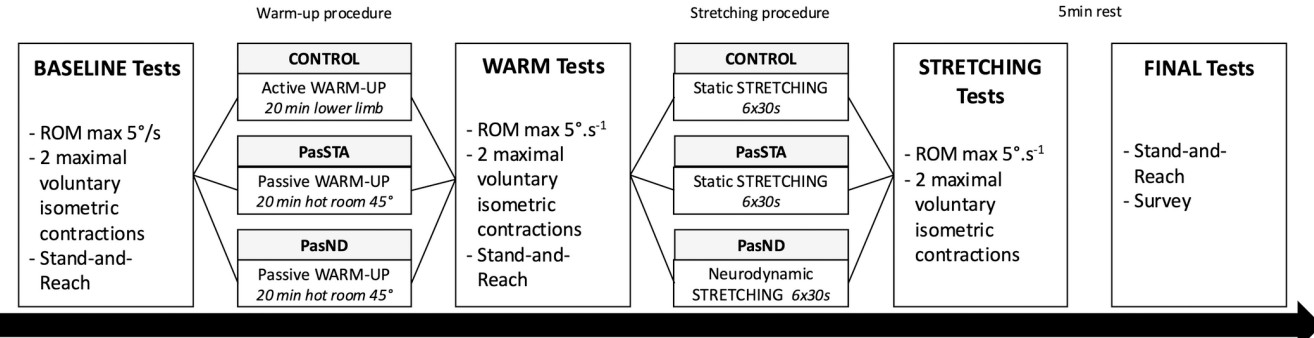

**Fig 1. General experimental design.**

stiffness and viscoelasticity, two maximal voluntary isometric contractions (MVC) with electromyography analysis (EMG), lower back and hamstrings flexibility assessed by the stand-and-reach test, and subjective rating of perceived exertion (RPE). The passive stretch and MVCs were conducted on the right side. All experimental sessions were performed on the same day of the week, at the same hour of the day and separated by seven days. The temperature of the laboratory was controlled and was statistically similar between all sessions (21.7 ± 1.5 °C). Volunteers were free to drink water ad libitum.

## Conditioning activities

**Passive warm-up.** It took place in a hot room at 45 °C [26]. Participants first lay supine for 10 minutes, followed by 10 minutes of sitting with their hips at approximately a 90° angle.

**Active warm-up.** It had a total duration of 20 minutes. It began with 10 minutes of cycling using an indoor ergocycle (CyclOps 400 Pro equipped with PowerTap, Madison, USA). The saddle and handlebar settings were adjusted individually. A constant power output was used (80 W and 100 W for women and men, respectively). Thereafter, submaximal contractions of the lower limb muscles were performed: 6 concentric knee extensions, 12 lunges, 10 concentric plantar flexions, 10 eccentric plantar flexions, 10 concentric knee flexions and 10 eccentric knee flexions. These contractions were followed by various athletic drills that included 20 rapid high knees and butt kicks, and five maximal counter movement jumps using the arms [27].

**Static stretching.** It was composed of 6 sets of 30 seconds and was performed with the participant lying supine on a massage table. The total stretch duration was therefore 180 seconds [16]. It involved a passive straight leg rise on the right side. Briefly, the experimenter placed a hand on the right knee and the other hand on the ipsilateral foot with the objective to maintain the knee extended and the ankle in dorsiflexion. The straight leg was slowly and progressively raised until reaching the maximum point of discomfort without experiencing pain. This position was maintained for 30 seconds, after which the lower limb was placed back on the table for 15 seconds.

**Neurodynamic nerve gliding.** This type of neural mobilization used almost the same configuration as static stretching. Participants were lying supine on a massage table with a slightly flexed cervical and thoracolumbar spine supported by a cushion as previously described [23]. The experimenter placed a hand to maintain the right knee extended and the other hand on the ipsilateral foot. The experimenter slowly and progressively raised the straight leg until the maximal discomfort endpoint without pain. Then, the participant simultaneously performed a cervical extension with dorsiflexion followed by a cervical flexion with plantar flexion. This cycle was repeated for 30 seconds and then there was a 15-second rest period with the lower limb positioned back on the table. This procedure was repeated six times.

## Tests

During the passive knee extension stretch, and MVCs, participants sat on the isokinetic dynamometer (System 5 Biodex Corporation, Shirley, NY, USA) as previously described [28]. Briefly, the left hip was maintained at a 100° angle, and the right hip was flexed at an angle of 45° between the right thigh and torso. The axis of the right knee (lateral condyle) was aligned with the axis of the dynamometer. Velcro straps were firmly attached distally on the volunteer's right thigh, above the right ankle, and over the pelvis, trunk, and contralateral thigh. The torque and EMG signals of the long head of biceps femoris (BF) and semitendinosus (ST) muscles were recorded using a Biopac MP150 system and the associated software (AcqKnowledge 4.2 for MP systems, Biopac System, Santa Barbara, CA, USA). A 1-kHz sampling frequency was used. Two pairs of silver chloride surface electrodes were placed over the BF and ST muscles following SENIAM recommendations. The interelectrode distance was 2 cm (center to center) and the reference electrode was fixed to the right patella. Low impedance (<5 kΩ) of the skin-electrode interface was obtained by shaving, abrading with sandpaper, and cleansing with alcohol. The EMG signals were amplified with a bandwidth frequency ranging from 10 to 500 Hz (common mode rejection ratio = 110 dB, gain = 500), and the root mean square value (RMS) was calculated over 250 ms sliding windows.

The passive knee extension stretch was used to evaluate the maximal knee range of motion, stiffness and viscoelasticity. It was conducted starting from a 90° knee joint angle until the point of maximum discomfort. A slow angular velocity (5 $°.s^{-1}$) was used to avoid the stretch reflex [17]. The maximal stretched position was held for 90 seconds, and then the leg was passively released to the initial position. During the initial phase of the passive stretch, the maximal passive torque and torque-time area were evaluated as stiffness indexes. While the leg was held in the stretched position, we assessed the torque decrease, *i.e.,* the difference between the greater and smaller torque values, as an index of viscoelasticity [29]. A constant angle procedure was used during each experimental session, *i.e.,* the PRE-test maximal range of motion was also used during the WARM and STRETCH tests.

The two isometric knee flexion MVCs were performed at a 90° knee angle. Participants were instructed to contract maximally for 5 seconds with a rest interval of the 15 seconds between the two attempts. The maximal knee flexion torque and the corresponding RMS values for both muscles were recorded. The best contraction, determined by the highest torque, was used for data analysis.

The lower back and hamstrings flexibility was assessed by the stand-and-reach test. This test was chosen in order to have a test that could easily be applied in the real practice. Briefly, participants stood upright with their feet parallel, and then they were requested to bend forward as far as possible [30]. The upper limbs were fully extended (elbow, wrist, and fingers), and the maximum distance achieved was quantified using a flexometer (TK500, Takei scientific instruments, Niigata, Japan). This procedure was performed once at each time point.

Rating of perceived exertion (RPE) was evaluated after warm-up and stretching procedures using a 10-points visual analogic scale (with 1 corresponding to a very light exercise and 10 a very hard exercise) [31].

## Statistical analyses

Values were expressed as mean ± standard deviation (SD). The normality and sphericity of the data were tested and confirmed by the Shapiro-Wilk and Mauchly's tests. Then, a two-way analysis of variance (ANOVA) with repeated measurements was performed (Condition × Time) with a Greenhouse-Geisser correction applied in the case of non-sphericity. The condition factor was the three experimental conditions (PasSTA, PasND and CON), and the time was the PRE, WARM, and STRETCH (for RPE assessed only after WARM and STRETCH). The Bonferroni post-hoc test was conducted in case of main condition or time effects, or time × condition interaction. The partial eta squared ($p\eta^2$) was calculated from ANOVA results, with values of 0.01, 0.06, and above 0.14 representing small, medium, and large differences, respectively. $p < 0.05$ was taken as the level of statistical significance for all comparisons. All statistical procedures were conducted using JASP (version 0.14, JASP Team 2020, University of Amsterdam, available free at https://jasp-stats.org/download/).

## Results

No main condition effect or condition × time interaction was obtained for MVC, BF RMS and ST RMS ($p > 0.05$, Table 1). In contrast, significant time effects were observed for ST RMS ($p = 0.004$). Post-hoc analyses revealed that the ST RMS (Fig 2) was significantly lower after STRETCH than WARM ($p = 0.026$).

During the passive stretch, significant time effect was observed for viscoelasticity ($p = 0.028$) (Table 1). Post-hoc analyses revealed a significant difference between PRE and WARM ($p = 0.025$). No difference was observed between PRE and WARM or WARM and STRETCH ($P < 0.05$). No significant main effects or interactions were obtained for passive torque and torque-time area ($p > 0.05$). During the stand-and-reach test, no main condition effect or condition × time interactions were observed ($p > 0.05$, Table 1). In contrast, a significant time effect was obtained ($p < 0.001$). The post-hoc analyses revealed a progressively increasing flexibility over time (PRE < WARM < STRETCH, $p = 0.001$, Fig 3).

RPE showed significant effects for condition and time and condition × time interaction (Table 1). RPE was significantly lower ($p < 0.001$) immediately after passive warm-up (1.2 ± 1.7 and 1.3 ± 1.4 for PasSTA and PasND, respectively) as compared to the control condition with active warm-up (4.5 ± 1.7 for CON) and after stretching (3.7 ± 1.6, 3.7 ± 1.9 and, 3.5 ± 1.8 for PasSTA, PasND and CON respectively).

**Table 1. Results of the two-way repeated measures ANOVA.**

| Variable | Effect | F | p | pη² |
|---|---|---|---|---|
| Maximal torque | Condition | 0.523 | 0.598 | 0.034 |
| | Time | 1.662 | 0.207 | 0.100 |
| | Condition × Time | 0.547 | 0.631 | 0.035 |
| BF RMS | Condition | 1.045 | 0.364 | 0.065 |
| | Time | 1.018 | 0.373 | 0.064 |
| | Condition × Time | 1.726 | 0.156 | 0.103 |
| ST RMS | Condition | 0.265 | 0.769 | 0.017 |
| | Time | 6.755 | 0.004 * | 0.310 |
| | Condition × Time | 2.163 | 0.084 | 0.126 |
| Passive Torque | Condition | 1.640 | 0.212 | 0.105 |
| | Time | 1.922 | 0.165 | 0.121 |
| | Condition × Time | 0.215 | 0.929 | 0.015 |
| Torque-Time Area | Condition | 1.297 | 0.289 | 0.085 |
| | Time | 2.490 | 0.101 | 0.151 |
| | Condition × Time | 1.912 | 0.121 | 0.120 |
| Viscoelasticity | Condition | 0.074 | 0.929 | 0.005 |
| | Time | 4.064 | 0.047* | 0.225 |
| | Condition x Time | 1.662 | 0.172 | 0.106 |
| Stand-and-Reach | Condition | 0.672 | 0.518 | 0.043 |
| | Time | 47.459 | < 0.001 * | 0.760 |
| | Condition × Time | 1.132 | 0.350 | 0.070 |
| RPE | Condition | 10.979 | < 0.001 * | 0.423 |
| | Time | 9.528 | < 0.001 * | 0.388 |
| | Condition × Time | 13.433 | < 0.001 * | 0.472 |

RMS: Root mean square; BF: Biceps femoris; ST: Semitendinosus; RPE: rate of perceived exertion. *: $p < 0.05$.

## Discussion

The aim of this study was to compare active and passive warm-up strategies combined with stretching on neuromuscular properties, flexibility and stiffness. Results confirmed our hypothesis since no difference was obtained between active and passive warm-up for maximal voluntary contractions and flexibility. Passive warm-up improved flexibility in a similar manner than active warm-up. In contrast, although flexibility gains were obtained after warm-up, adding stretching, irrespective of the stretching modality applied, further increased this parameter. From a practical perspective, these findings suggest the practitioner may consider passive warm-up as an alternative to active warm-ups, given its advantage in reducing energy expenditure and fatigue [12]. Stretching, when used in conjunction with a warm-up routine, remains an effective strategy for further enhancing flexibility, regardless of the technique employed.

### Effects of warm-up

The present study first revealed increases in stand-and-reach following both the passive and active warm-up strategies. Studies have previously reported the efficiency of passive strategies on range of motion using local hot packs or whole-body hot water baths [32,33]. Active warm-up strategies such as jogging, cycling and/or athletic drills have also shown effective for range of motion improvements [4,5,34]. These improvements have been suggested to be mediated via muscle or core temperature-related mechanisms [5,10,33] which would affect the neuromuscular system. Amongst these,

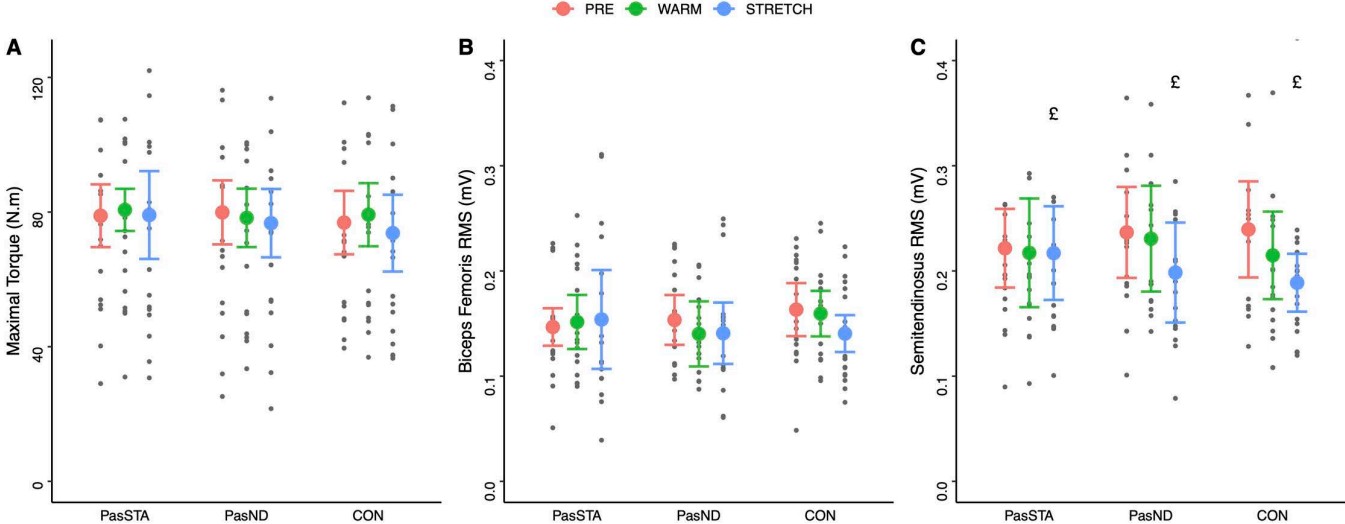

**Fig 2. Maximal voluntary torque (A), biceps femoris RMS (B), and semitendinosus RMS (C) before the experimental conditions (PRE), after warm-up (WARM) and after stretching (STRETCH) for all conditions. PasSTA: passive warm-up followed by static stretching; PasND: passive warm-up followed by neurodynamic stretching and CON: active warm-up followed by static stretching.** Values are means ± SD. £: significant differences with WARM (p < 0.05).

stretching tolerance (pain perception) and changes in muscle passive properties (passive stiffness and viscosity) are the most likely mechanisms [5,10,35]. Our results partially refuted some of these mechanisms. Indeed, there were no significant changes in the passive torque, torque-time area, and viscoelasticity during the passive knee extension stretch test. However, these findings partly supported previous studies [32]. In a similar study comparing passive and active warm-up strategies, authors found an increase in range of motion without any alteration of passive muscle properties [32]. Accordingly, these authors concluded that the gains in range of motion were primarily due to an increase in stretch tolerance as evidenced by higher passive torque for greater range of motion. While our study utilized a constant-angle procedure for measuring passive properties, our results support the likely role of stretch tolerance in increasing flexibility.

Warm-up, whether it is conducted with passive [36] or active strategies [10,27], are generally associated with improvements of the contractile function. Several mechanisms are implied such as calcium kinetics or muscle fluids [36]. In the present study, the contractile function was measured via a maximal voluntary contraction. No statistical differences were obtained for the maximal torque or corresponding EMG activity. These controversial results under passive conditions could firstly be attributed to the technical procedures used for warm-up. Studies generally employed hot-water immersions. Although, a hot room, specifically a sauna, was used here, the temperature applied (45 °C) falls within the temperature ranges generally recommended (for safety or tolerance) [36]. In contrast, the 20 minutes heat administration was shorter than many other studies. For example, some research observed improved contractile function following 90 minutes immersion in hot-water [37]. It is reasonable to speculate that a longer heat administration would have increased force production capacity. However, one can question the applicability of such long durations in sports settings.

The lack of increase in contractile function after active warm-up can secondly be explained by several main hypotheses. The contractile function was evaluated using maximal voluntary isometric contractions. However, this condition was not included in our warm-up set-up. The well-known specificity of warm-up effects (related to motor learning) should therefore be respected to observe positive effects after a comprehensive warm-up [27]. Finally, the test and data analyses could also explain the lack of contractile function enhancement. Applying different tests (such as a direct measurement of the rate of force development) or other analyses might be of interest [7,36]. Indeed, temperature-dependent effects

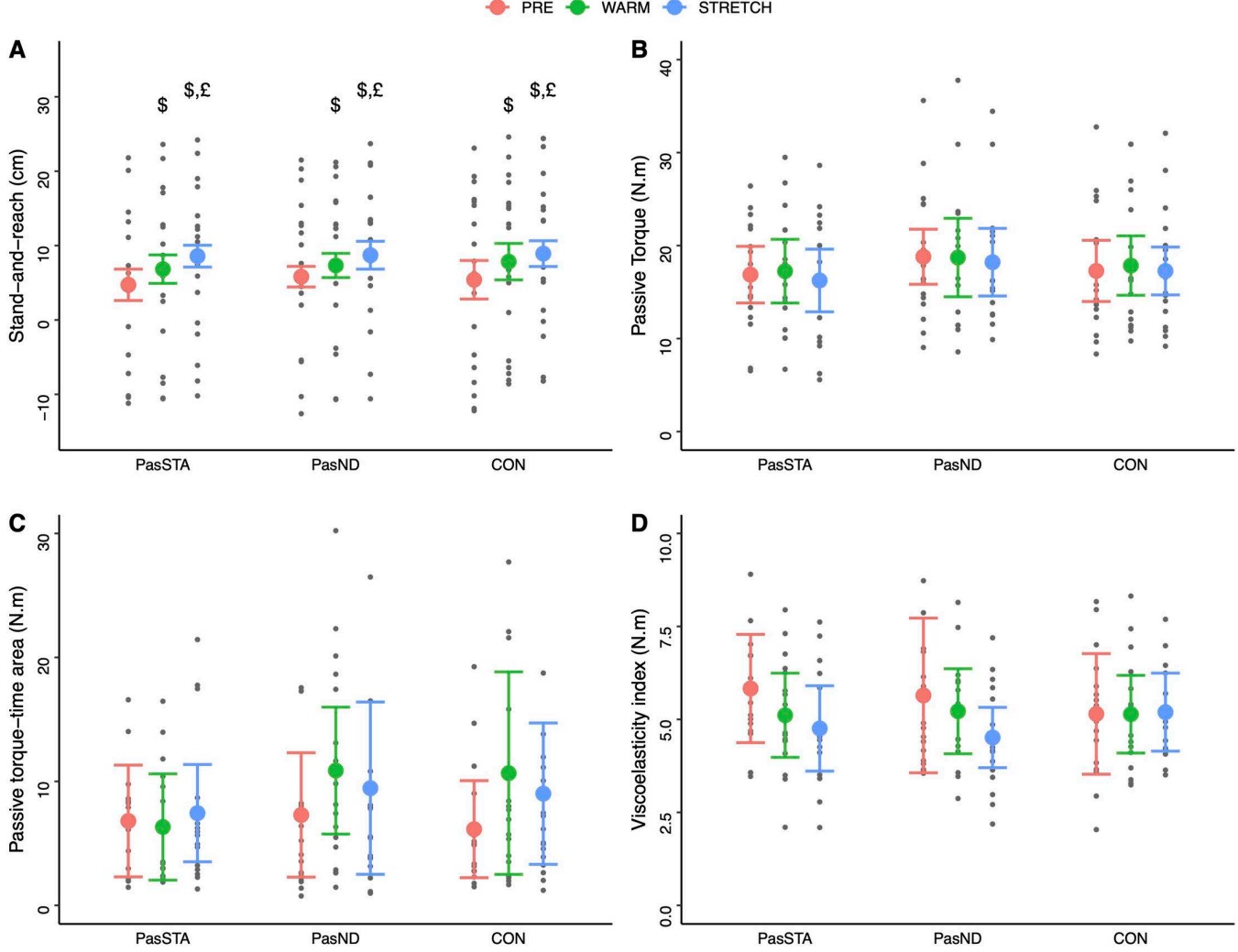

**Fig 3. Stand-and-reach (A), passive torque (B), torque-time area (C), viscoelasticity index (D), and RPE (D) before the experimental conditions (PRE), after warm-up (WARM) and after stretching (STRETCH) for all conditions. PasSTA: passive warm-up followed by static stretching; PasND: passive warm-up followed by neurodynamic stretching and CON: active warm-up followed by static stretching.** Values are means ± SD. $: Significant differences with PRE (p < 0.05). £: significant differences with WARM (p < 0.05).

appeared to be more effective on fast contractions [37]. However, one should acknowledge the lack of the skin or muscle temperature control. Indeed, it would have been useful to control the likely alterations in muscle temperature after both warming procedures and stretching.

## Stretching effects

After passive or active warm-up, 6 sets of 30 seconds of static or neurodynamic stretching were applied. In contrast to our a priori hypothesis, an exacerbated increase in flexibility was observed for both stretching modalities. Such finding is partly in contradiction with previous studies [34]. Indeed, these authors concluded that 3 sets of 10 seconds stretching within a full, dynamic warm-up had no effect on flexibility. Accordingly, the conflicting findings might be related to the duration proposed for stretching. Although a short-volume of static stretching may slightly enhance flexibility, it has been reported

that flexibility improvements from high-volume is better than short-volume of static stretching [38]. Indeed, authors did not exclude the possibility that adding longer static stretching periods instead of short ones to an active and full warm-up could enhance flexibility [39].

However, as for warm-up effects, this increase in flexibility was not associated with alterations in most muscle passive properties. This result could appear surprising since numerous studies have registered changes in muscle-tendon unit stiffness and viscosity after similar duration stretches [17,28,40]. However, changes in passive torque may not always be detected and could be influenced by the stretching technique and by measurements of passive properties. For instance, the impact of stretching on passive torque is typically observed during the final degrees of a passive stretch procedure [18]. The constant-angle procedure utilized in our study might obscure such changes. Furthermore, there was a large effect size on passive torque that approached statistical significance. It is probable that incorporating more participants or employing an alternative procedure would have revealed alterations in passive properties.

In contrast, the neural drive, witnessed by the ST RMS, was impaired after both stretching modalities. These results therefore suggested that conducting moderate duration stretching in association with a comprehensive warm-up was inefficient for subsequent performance [4]. Such results are commonly obtained in the literature [14,15,41] and point towards numerous central or peripheral origins that are regularly explored in research studies. However, despite the decrease in ST RMS, no associated alteration of the maximal voluntary torque was observed. Such finding is in contradiction with most of the literature concluding that prolonged stretching significantly impaired the maximal torque production capacity [14–16,38]. One should remember that both women and men were included in this study. It will inherently conduct to large maximal voluntary torque variability that can mask a potential time effect.

Interestingly, no difference was observed between static and neurodynamic nerve gliding procedures. This result suggested that these two stretching procedures could be used interchangeably for flexibility increases. However, both stretching procedures imply different mechanisms since different tissues are involved. While static stretching mostly alters the neuromuscular and muscle-tendon systems [17,40] neurodynamic procedures additionally affect nerves stiffness [42]. A decrease in nerve stiffness has been registered immediately after neurodynamic stretches [42]. However, the exact effects of neurodynamic techniques (whether using sliding or tensioning) need further studies to better determine the subsequent effects and mechanisms.

## Limitation and perspectives

The present study has several limitations that should be acknowledged. As previously stated, the skin or muscle temperature has not been controlled. In addition, the stand-and-reach test was used as a flexibility measurement. The rationale for using this test is questionable since it involved both the lower back and hamstring muscles while stretching only engaged hamstring muscles. The conclusions obtained during this flexibility test and the passive knee extension test should therefore be compared with caution. However, these muscles were involved in our active and passive warm-up routines, which corresponded to the primary aim of our study. Finally, a moderate stretch duration, generally not recommended during warm-ups, was used here. Such rather long stretch duration was chosen to exacerbate stretching effects since less in known in the literature for the neurodynamic nerve gliding stretching modality.

Beside these limitations, the present study provided important practical applications for athletes. Firstly, passive strategies could be applied within warm-up routines. In addition to the gain in flexibility, passive warm-up has multiple interesting advantages for instance while reducing energy consumption. RPE was largely reduced with passive procedure as compared to active ones. Moreover, passive strategies permit to gain time during the whole pre-activity routine for instance to conduct concomitant mental tasks, such as mental imagery. Secondly, both warm-up strategies and stretching induced the same effects on muscle flexibility and strength. Although different neuromuscular mechanisms are likely involved, these various techniques can be used interchangeably to mitigate training monotony. However, both stretching modalities further increase flexibility but may impair subsequent muscle performance. Therefore, practitioners

should prescribe stretching exercises based on the specific sport, event, and key performance indicators. Further studies should be conducted to better determine neuromuscular mechanisms and assess the utility of neurodynamic techniques in sports science.

## Supporting information

**S1 File. Inclusivity in global research.**
(DOCX)

## Acknowledgements

The authors would like to thank the technical support from the Centre d'Expertise de la Performance from the Université Bourgogne Europe. We thank all participants for their contribution and dedication to the study.

## Author contributions

**Conceptualization:** Marion Hitier, Denis César Leite Vieira, Carole Cometti, Joao Luiz Quagliotti Durigan, Nicolas Babault.

**Formal analysis:** Denis César Leite Vieira.

**Funding acquisition:** Nicolas Babault.

**Investigation:** Marion Hitier, Denis César Leite Vieira, Carole Cometti.

**Methodology:** Marion Hitier, Joao Luiz Quagliotti Durigan, Nicolas Babault.

**Supervision:** Nicolas Babault.

**Validation:** Joao Luiz Quagliotti Durigan, Nicolas Babault.

**Visualization:** Carole Cometti, Joao Luiz Quagliotti Durigan.

**Writing – original draft:** Marion Hitier, Nicolas Babault.

**Writing – review & editing:** Marion Hitier, Denis César Leite Vieira, Carole Cometti, Joao Luiz Quagliotti Durigan, Nicolas Babault.

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
