## [Decision Letter · Decision Letter 0]

11 Mar 2025

PONE-D-25-06600Passive vs. active warm-up combined with stretching on hamstring flexibility and maximal voluntary contractionsPLOS ONE

Dear Dr. Babault,

Thank you for submitting your manuscript to PLOS ONE. After careful consideration, we feel that it has merit but does not fully meet PLOS ONE’s publication criteria as it currently stands. Therefore, we invite you to submit a revised version of the manuscript that addresses the points raised during the review process.

We look forward to receiving your revised manuscript.

Kind regards,

Masatoshi Nakamura, Ph.D.

Academic Editor

PLOS ONE

Journal Requirements:

“Région Bourgogne Franche-Comté (2020Y-22065 and 2022Y-13186)

National Council for Scientific and Technological Development–CNPq (200391/2022-4)”

“The authors would like to thank the financial support from the Centre d’Expertise de la Performance from the Université de Bourgogne, the Région Bourgogne Franche-Comté (2020Y-22065 and 2022Y-13186), and the National Council for Scientific and Technological Development–CNPq (200391/2022-4). We thank all participants for their contribution and dedication to the study.”

“Région Bourgogne Franche-Comté (2020Y-22065 and 2022Y-13186)

National Council for Scientific and Technological Development–CNPq (200391/2022-4)”

5. In the online submission form, you indicated that “All the data generated and analysed as part of this study are not accessible to the public but are available on request.”

Reviewers' comments:

Reviewer's Responses to Questions

**Comments to the Author**

1. Is the manuscript technically sound, and do the data support the conclusions?

Reviewer #1: Yes

Reviewer #2: Partly

2. Has the statistical analysis been performed appropriately and rigorously?

Reviewer #1: Yes

Reviewer #2: Yes

3. Have the authors made all data underlying the findings in their manuscript fully available?

Reviewer #1: Yes

Reviewer #2: Yes

4. Is the manuscript presented in an intelligible fashion and written in standard English?

Reviewer #1: Yes

Reviewer #2: Yes

5. Review Comments to the Author

Reviewer #1: The English writing is good but there are a number of minor grammatical problems that should be fixed. I do not have time to identify all grammatical problems. If accepted hopefully the copy editors will help with that.

Abstract

Line 41: “consisted of” not “consisted in”

Introduction

Line 69-70: The phrase “multiple neural alterations” is quite obtuse. Please be more specific.

Line 79-80: Static stretching in general does not have detrimental effects on force production. It has been reported in a number of reviews that it is “prolonged” static stretching (typically more than 60-seconds per muscle group) that can induce performance impairments. (Behm and Chaouachi 2011, Kay and Blazevich 2012, Behm, Blazevich et al. 2016, Chaabene, Behm et al. 2019, Behm, Kay et al. 2021)

Methods

Participants

Obviously, males and females are generally dichotomous in their anthropometric characteristics so providing the reader with one mean value will describe some hybrid male / female subject. You need to provide separate descriptions for each sex.

Line 123, 131, 201: Stand and reach test would provide an index of lower back and hamstrings flexibility but it would not be global. How does this test measure shoulder, neck flexibility for example? That type of test would be global. Please remove this term from the manuscript.

Discussion

Line 345: Authors suggest this is a common result in the literature which is correct. Would it not be stronger to then cite reviews rather than two single original research papers. The review references provided at the end of this critique would be stronger evidence.

Line 351: It seems the only difference between your static stretching and neurodynamic stretching procedures was to flex and extend the neck while also performing dorsiflexion. I would agree that this action would have a greater emphasis on nerve stretching but the supine position with hip flexion and an extended knee would still provide tension on the musculotendinous structures. So I feel it is false to say it primarily affects nerve stiffness. It affects both!

Suggested references

Behm, D. G., A. J. Blazevich, A. D. Kay and M. McHugh (2016). "Acute effects of muscle stretching on physical performance, range of motion, and injury incidence in healthy active individuals: a systematic review." Appl Physiol Nutr Metab 41(1): 1-11.

Behm, D. G. and A. Chaouachi (2011). "A review of the acute effects of static and dynamic stretching on performance." Eur J Appl Physiol 111(11): 2633-2651.

Behm, D. G., A. D. Kay, G. S. Trajano and A. J. Blazevich (2021). "Mechanisms underlying performance impairments following prolonged static stretching without a comprehensive warm-up." Eur J Appl Physiol 121(1): 67-94.

Chaabene, H., D. G. Behm, Y. Negra and U. Granacher (2019). "Acute Effects of Static Stretching on Muscle Strength and Power: An Attempt to Clarify Previous Caveats." Front Physiol 10: 1468.

Kay, A. D. and A. J. Blazevich (2012). "Effect of acute static stretch on maximal muscle performance: a systematic review." Med Sci Sports Exerc 44(1): 154-164.

Reviewer #2: Thank you for the opportunity to review this manuscript. The authors examined the combined effects of passive warm-up and stretching. There is a lack of discussion on stretching. Also, sit and reach were used as a measure of flexibility, but comparing them to changes in the stiffness of the hamstrings is questionable.

Abstract

Line 46-47: When was EMG measured? During the flexibility measurement?

Line 47: What does “flexibility” indicate (maybe range of motion)? Authors should state specifically throughout the manuscript. Did the authors use the term "Range of motion" and "Flexibility" interchangeably?

Line 53: If stiffness is measured, please describe it in the abstract method.

Introduction

Neurodynamic stretching is not a common type of stretching. Please explain in the introduction.

Materials and Methods

Line 115: Why did you choose the effect size of 0.35? Please explain the rationale.

Line 120: Please explain the specific blinding procedure.

Why was the stand-and-reach test used? This test includes the flexibility of the muscles other than the hamstrings (e.g., the trunk muscle). But, stretching was performed only on the hamstrings.

The room temperature of 45 degrees is high. Did the authors regulate water intake of participants?

Line 144: This study used six sets of 30 seconds of stretching. Previous studies have shown that more than 30 seconds lowers performance (Behm et al., 2011). Why did the authors use this stretching?

Results

Line 243: Please explain the reason for the discrepancy between the main effect for time and the post hoc analysis.

Discussion

Line 278-294 This study examined the combined effects of passive warm-up and stretching on range of motion and stiffness. The author describes the possibility that the changes in sit and reach may be related to tolerance, but I have a question about that because sit and reach includes not only the hamstring but also the trunk flexibility.

Line 327 Reference 37 study did not use static stretching.

Previous studies have shown that static stretching for more than 3 minutes decreases stiffness (Matsuo et al., JSCR 2013; Takeuchi et al., JSSM, 2023). Muscle strength is also decreased by stretching for more than 30 seconds after warm-up (Behm 2011). Why did muscle strength not decrease despite a decrease in RMS? Of course, the results of this study are not necessarily the same as in previous studies, but the reasons why the results differ from many previous studies should be discussed in more detail.

Please describe the limitations of this study.

6. PLOS authors have the option to publish the peer review history of their article (what does this mean? ). If published, this will include your full peer review and any attached files.

**Do you want your identity to be public for this peer review?** For information about this choice, including consent withdrawal, please see our Privacy Policy .

Reviewer #1: **Yes: ** David G Behm

Reviewer #2: No

---

## [Author Response · Author response to Decision Letter 1]

17 Mar 2025

Response to Reviewer #1

The English writing is good but there are a number of minor grammatical problems that should be fixed. I do not have time to identify all grammatical problems. If accepted hopefully the copy editors will help with that.

Response: Authors first thank the reviewer for the interesting and helpful comments. Please see below a point-by-point response to all of your comments. Alterations in the manuscript are presented in red font. Authors have carefully read and corrected the manuscript to limit grammatical problems.

Abstract

Line 41: “consisted of” not “consisted in”

Response: Corrected

Introduction

Line 69-70: The phrase “multiple neural alterations” is quite obtuse. Please be more specific.

Response: Some specific alterations have been given.

Line 79-80: Static stretching in general does not have detrimental effects on force production. It has been reported in a number of reviews that it is “prolonged” static stretching (typically more than 60-seconds per muscle group) that can induce performance impairments. (Behm and Chaouachi 2011, Kay and Blazevich 2012, Behm, Blazevich et al. 2016, Chaabene, Behm et al. 2019, Behm, Kay et al. 2021)

Response: Of course, authors agree with the reviewer comment. This sentence has been modified to make it clear for the reader.

Methods

Participants

Obviously, males and females are generally dichotomous in their anthropometric characteristics so providing the reader with one mean value will describe some hybrid male / female subject. You need to provide separate descriptions for each sex.

Response: Anthropometric description has been separated between women and men.

Line 123, 131, 201: Stand and reach test would provide an index of lower back and hamstrings flexibility but it would not be global. How does this test measure shoulder, neck flexibility for example? That type of test would be global. Please remove this term from the manuscript.

Response: The reviewer is right. We corrected this erroneous term. This term has also been removed in discussion.

Discussion

Line 345: Authors suggest this is a common result in the literature which is correct. Would it not be stronger to then cite reviews rather than two single original research papers. The review references provided at the end of this critique would be stronger evidence.

Response: As suggested by the reviewer, we changed references and only included review from your suggestions.

Line 351: It seems the only difference between your static stretching and neurodynamic stretching procedures was to flex and extend the neck while also performing dorsiflexion. I would agree that this action would have a greater emphasis on nerve stretching but the supine position with hip flexion and an extended knee would still provide tension on the musculotendinous structures. So I feel it is false to say it primarily affects nerve stiffness. It affects both!

Response: We agree with your comment and altered the part of the discussion.

Suggested references

Behm, D. G., A. J. Blazevich, A. D. Kay and M. McHugh (2016). "Acute effects of muscle stretching on physical performance, range of motion, and injury incidence in healthy active individuals: a systematic review." Appl Physiol Nutr Metab 41(1): 1-11.

Behm, D. G. and A. Chaouachi (2011). "A review of the acute effects of static and dynamic stretching on performance." Eur J Appl Physiol 111(11): 2633-2651.

Behm, D. G., A. D. Kay, G. S. Trajano and A. J. Blazevich (2021). "Mechanisms underlying performance impairments following prolonged static stretching without a comprehensive warm-up." Eur J Appl Physiol 121(1): 67-94.

Chaabene, H., D. G. Behm, Y. Negra and U. Granacher (2019). "Acute Effects of Static Stretching on Muscle Strength and Power: An Attempt to Clarify Previous Caveats." Front Physiol 10: 1468.

Kay, A. D. and A. J. Blazevich (2012). "Effect of acute static stretch on maximal muscle performance: a systematic review." Med Sci Sports Exerc 44(1): 154-164.

Response to Reviewer #2

Thank you for the opportunity to review this manuscript. The authors examined the combined effects of passive warm-up and stretching. There is a lack of discussion on stretching. Also, sit and reach were used as a measure of flexibility, but comparing them to changes in the stiffness of the hamstrings is questionable.

Response: Authors thank the reviewer for the interesting and helpful comments. Please see below a point-by-point response to all of your comments. Alterations in the manuscript are presented in red font.

Abstract

Line 46-47: When was EMG measured? During the flexibility measurement?

Response: EMG was recorded during the maximal voluntary contractions. It has been added.

Line 47: What does “flexibility” indicate (maybe range of motion)? Authors should state specifically throughout the manuscript. Did the authors use the term "Range of motion" and "Flexibility" interchangeably?

Response: Throughout the manuscript we used these two different terms. They are not used interchangeably. Flexibility referred to some practical/functional tests such as the stand and reach used here. Range of motion referred to lab tests such as the passive knee extension used here. Authors have tried to better define terms and make it clear throughout the manuscript.

Line 53: If stiffness is measured, please describe it in the abstract method.

Response: Due to space restrictions, it was difficult to describe the stiffness assessment. However, we added it was evaluated during the passive knee extension test.

Introduction

Neurodynamic stretching is not a common type of stretching. Please explain in the introduction.

Response: As recommended, we added a sentence to indicate it is generally used in clinical settings.

Materials and Methods

Line 115: Why did you choose the effect size of 0.35? Please explain the rationale.

Response: Sorry it was a slight error. We did the a priori analysis in order to obtain a large effet (f = 0.40) with a power of 0.9. We corrected this mistake.

Line 120: Please explain the specific blinding procedure.

Response: We use and single-blind procedure. Both the experimenter and the volunteers knew the experimental conditions. The different data were collected, and files were coded. Blinded data were thereafter analyzed by one of the authors. Statistics were also conducted while being blinded. Coding was finally revealed after all statistics were done. No modification was done in the manuscript since authors think indicating data analyses and statistics were conducted blinded was sufficient enough.

Why was the stand-and-reach test used? This test includes the flexibility of the muscles other than the hamstrings (e.g., the trunk muscle). But, stretching was performed only on the hamstrings.

Response: Authors acknowledge, more muscle groups are considered. However, we decided to include this test to have a more "functional" result. By "functional", authors mean a test that could easily be applied in a real practice. This outcome is easily comprehensible by the practitioner. A sentence has been added in methods for clarity ('tests' part).

The room temperature of 45 degrees is high. Did the authors regulate water intake of participants?

Response: No regulation of water intake was made. Volunteers were free to drink as much water as they wanted. We added a sentence in the 'experimental design' paragraph.

Line 144: This study used six sets of 30 seconds of stretching. Previous studies have shown that more than 30 seconds lowers performance (Behm et al., 2011). Why did the authors use this stretching?

Response: Authors obviously know that prolonged static stretch could be detrimental for subsequent performance (e.g., force). We decided to apply such long stretch for the comparison with the neurodynamic nerve gliding procedure. In fact, this form of stretching scarcely studied is sports settings. We decided to have a rather long stretch to determine whether differences could be obtained between stretching procedures. The prolonged stretch duration has been acknowledged in discussion.

Results

Line 243: Please explain the reason for the discrepancy between the main effect for time and the post hoc analysis.

Response: The authors thank the reviewer for this comment. After a careful new check of our values, we noticed some small mistakes. First, we corrected two p-values in table 1. We forgot to take into account the sphericity correction for the ANOVA p-values. It was corrected for the maximal torque interaction (0.631 instead of 0.702) and for the time effect of the viscoelasticity (0.047 instead of 0.028). These corrections do not change our results. The other values have been double checked to avoid any other mistake. But the reviewer is right, a significant difference was obtained for post-hoc pairwise comparisons. No difference was obtained between PRE and WARM (P = 0.358) or between WARM and STRETCH (P = 0.681). However, a difference was obtained between PRE and STRETCH ((P = 0.025). The results section has been corrected accordingly

Discussion

Line 278-294 This study examined the combined effects of passive warm-up and stretching on range of motion and stiffness. The author describes the possibility that the changes in sit and reach may be related to tolerance, but I have a question about that because sit and reach includes not only the hamstring but also the trunk flexibility.

Response: The reviewer is right. As indicated previously, authors wanted a more functional / practical test that involved numerous muscle groups. This limitation is acknowledged at the end of the discussion.

Line 327 Reference 37 study did not use static stretching.

Response: Ref 37 investigated static stretching. However, the reviewer is right for ref 31. (ref 33 in the revised manuscript). Accordingly, authors deleted this reference from this sentence.

Previous studies have shown that static stretching for more than 3 minutes decreases stiffness (Matsuo et al., JSCR 2013; Takeuchi et al., JSSM, 2023). Muscle strength is also decreased by stretching for more than 30 seconds after warm-up (Behm 2011). Why did muscle strength not decrease despite a decrease in RMS? Of course, the results of this study are not necessarily the same as in previous studies, but the reasons why the results differ from many previous studies should be discussed in more detail.

Response: The lack of torque alteration was effectively uncommon in the literature. We tried to discuss such results by considering both sexes were included in the study. The large variability seen in fig 2A could be one likely explanation. We tried to include sex in the statistical analyses. The p-value of the 3-way ANOVA was close to 0.1 but remain unsignificant. We also did some new analyses with the sex effect for all variables. No main sex effect nor interactions were observed.

Please describe the limitations of this study.

Response: A limitation section has been added in 'discusion' section.

---

## [Decision Letter · Decision Letter 1]

24 Mar 2025

Passive vs. active warm-up combined with stretching on hamstring flexibility and maximal voluntary contractions

PONE-D-25-06600R1

Dear Dr. Babault,

We’re pleased to inform you that your manuscript has been judged scientifically suitable for publication and will be formally accepted for publication once it meets all outstanding technical requirements.

Kind regards,

Masatoshi Nakamura, Ph.D.

Academic Editor

PLOS ONE

Additional Editor Comments (optional):

Reviewers' comments:

Reviewer's Responses to Questions

**Comments to the Author**

1. If the authors have adequately addressed your comments raised in a previous round of review and you feel that this manuscript is now acceptable for publication, you may indicate that here to bypass the “Comments to the Author” section, enter your conflict of interest statement in the “Confidential to Editor” section, and submit your "Accept" recommendation.

Reviewer #1: All comments have been addressed

Reviewer #2: All comments have been addressed

2. Is the manuscript technically sound, and do the data support the conclusions?

Reviewer #1: Yes

Reviewer #2: (No Response)

3. Has the statistical analysis been performed appropriately and rigorously?

Reviewer #1: Yes

Reviewer #2: (No Response)

4. Have the authors made all data underlying the findings in their manuscript fully available?

Reviewer #1: Yes

Reviewer #2: (No Response)

5. Is the manuscript presented in an intelligible fashion and written in standard English?

Reviewer #1: Yes

Reviewer #2: (No Response)

6. Review Comments to the Author

Reviewer #1: Only two minor comments

1. p.6: body mass 2were 0.8 ± 1.7 years,: Delete "2"

2. As you know there are three muscles in the hamstrings group. Hence just like you would use the term quadriceps (plural: 4 muscles, almost nobody would ever say quadricep), the term should be hamstrings (plural: three muscles) or hamstring muscles (the term muscle is plural).

Other than that the authors have adequately addressed my concerns.

Reviewer #2: (No Response)

7. PLOS authors have the option to publish the peer review history of their article (what does this mean? ). If published, this will include your full peer review and any attached files.

**Do you want your identity to be public for this peer review?** For information about this choice, including consent withdrawal, please see our Privacy Policy .

Reviewer #1: **Yes: ** David G. Behm

Reviewer #2: No

---

## [Editor Report · Acceptance letter]

PONE-D-25-06600R1

PLOS ONE

Dear Dr. Babault,

I'm pleased to inform you that your manuscript has been deemed suitable for publication in PLOS ONE. Congratulations! Your manuscript is now being handed over to our production team.

Kind regards,

on behalf of

Dr. Masatoshi Nakamura

Academic Editor

PLOS ONE